# Sex and pressure effects of foam rolling on acute range of motion in the hamstring muscles

**Norikazu Hirose**[1]*, **Akane Yoshimura**[2], **Kei Akiyama**[1], **Atsuya Furusho**[3]

1 Faculty of Sport Sciences, Waseda University, Tokyo, Japan, 2 Faculty of Education and Integrated Arts and Sciences, Waseda University, Tokyo, Japan, 3 Graduate School of Sport Sciences, Waseda University, Saitama, Japan

* toitsu_hirose@waseda.jp

## Abstract

This study investigated the effects of pressure levels and sex on acute range of motion (ROM) changes during foam rolling (FR) using a soccer ball. Twenty collegiate athletes (10 males, 10 females) performed FR on their hamstrings at low (15–25% body weight) or high (45–55%) pressure levels for 2 minutes. ROM was assessed through passive straight leg raise (PSLR) and passive knee extension (PKE) tests before, immediately after, and 10 minutes post-intervention. Results showed that FR significantly improved hip and knee ROM across all conditions, with benefits persisting for at least 10 minutes. No notable differences were observed between pressure levels, and sex did not affect the magnitude of improvement. Perceived pain during FR did not significantly influence ROM outcomes. These findings demonstrate that FR using a simple tool like a soccer ball is beneficial for improving ROM in both males and females, regardless of pressure intensity. This suggests that FR provides a practical, accessible method for improving ROM in athletes and other populations.

## Introduction

Improving the range of motion (ROM) before any exercise in athletes and patients is a frequent goal in conditioning and rehabilitation, where foam rolling (FR) is widely used. FR is a beneficial technique that increases ROM immediately after intervention [1], without a reduced performance of jump and strength [2].

Many studies have explored the potential factors influencing the outcomes of FR to develop effective, individualized protocols. Key factors include the individual's characteristics, the targeted muscle group, and specific conditions of the FR intervention. For instance, ROM improvements are often more pronounced in the hamstrings and quadriceps compared to the triceps surae [3]. A recent meta-analysis found that studies including only male participants showed smaller changes in ROM after FR interventions compared to studies including females or mixed-sex samples. This suggests that ROM improvements induced by FR may be less pronounced in males. However, there is no consensus on this issue, highlighting the need for further empirical research to clarify these findings [1].

**Data availability statement:** All data supporting this study's findings are publicly available in the Dryad Digital Repository (DOI: https://doi.org/10.5061/dryad.1ns1rn93x).

**Funding:** The author(s) received no specific funding for this work.

**Competing interests:** The authors have declared that no competing interests exist.

Under specific FR conditions, other determinants, such as intervention speed [4], vibration [1], and duration [4] may also influence outcomes. Nakamura et al. reported that FR for over 90 seconds effectively increased ROM immediately without affecting muscle stiffness or strength [4]. FR intensity, modulated by factors such as applied pressure [5], contact area [6], rigidity [6], and equipment density [7,8], is also significant but remains poorly understood due to limited research. For example, altering applied pressure has inconsistent effects on acute ROM changes. While some studies found no significant impact of pressure changes on knee ROM after quadriceps FR [5], others observed similar results for ankle ROM after calf FR with equipment of varying densities [8]. These findings suggest that pressure variations may not universally affect ROM improvements, although this may differ across muscle groups, such as the hamstrings [1]. Further studies are warranted to explore pressure-dependent differences in acute ROM changes in specific muscles.

Transient ROM changes from FR interventions are attributed to both systemic and local factors. Systemic factors may include increased pain thresholds [9,10] and reduced sympathetic activity [11,12], leading to muscular relaxation. Local effects may involve enhanced blood flow [13,14], improved vascular function [15], and altered muscle stiffness [16,17]. However, the mechanisms underlying these changes are complex and not fully understood.

Applying rolling and pressure stimulation may activate mechanoreceptors like Ruffini and Pattini receptors in the skin's superficial layer, contributing to sympathetic inhibition [11]. Painful stimuli may also elicit diffuse noxious inhibitory controls (DNIC) or human-equivalent conditioned pain modulation, which attenuates distant responses [18–20]. While pain is not always a prerequisite for FR's effectiveness [21], the intensity of the mechanical stimuli may have a limited impact on outcomes [21,22]. This suggests that rolling and pressure stimulation are critical for acute ROM improvements, regardless of equipment type or intensity. By applying these stimuli to targeted muscles, athletes and patients can achieve ROM gains using readily available tools, such as a soccer ball.

The aim of this study is to investigate the acute effects of FR on ROM based on sex and pressure. Specifically, we examined the influence of FR intensity and sex on acute ROM changes by applying FR intervention to the hamstring muscles using a commercially available soccer ball. We hypothesized that FR applied to the hamstring muscles would improve ROM, with more pronounced acute changes in females and no significant pressure-dependent differences.

## Materials and methods

### Participants

Before recruitment, we calculated the minimum number of participants using G＊Power 3.1.9.4 (Heinrich Heine Universität Düsseldorf, Germany). The calculation for the a priori F test with analysis of variance (ANOVA) repeated measures, within-between interaction of six groups (two sexes and three pressures) and three measurements (three-time points), was conducted, given an effect size of 0.25, an α error probability of 0.05, and a power of 0.80, with a correlation among repeated measures of 0.5 and a non-sphericity correction (ε) of 1. The total required sample size was determined to be 30 for six groups, with five participants per group, which corresponds to five participants for each sex as the minimum sample size. However, to reduce the influence of individual differences and measurement errors, address uncertainties in effect size, mitigate the impact of missing data and adjustments for non-sphericity, and enhance the reproducibility of the experimental results, the study included 10 participants per group. This resulted in a total of 20 participants (10 males and 10 females). We recruited collegiate athletes who trained for at least 3 days per week. Our exclusion criteria were: current

injuries, a history of severe injury in the lower extremity or lower back, and muscle soreness prior to the experiment. The recruitment process was carried out from the 8th of April 2021 to the 29th of April 2021. A total of 20 athletes (10 female and 10 male) participated in this study (seven for lacrosse, five for soccer, three for hockey, three for ice hockey, and two for CrossFit; age, 20.4 ± 2.1 years; body height, 165.7 ± 7.7 cm; body weight, 62.9 ± 10.6 kg). They were asked not to engage in exhaustive activity within 24 h before the experiment and from any physical activity, including static stretching, beyond the activity of daily living within 12 h of the experiment. This study was conducted in full compliance with the Declaration of Helsinki. Ethical approval was obtained from the Ethics Committee of Waseda University, approval number 2020-418. All participants were provided comprehensive information regarding the study objectives, procedures, potential risks, and benefits. Written informed consent was obtained from each participant prior to their involvement in the study. Participants were informed that their participation was voluntary and that they could withdraw from the study at any time without any repercussions. Furthermore, all data collected were anonymized to ensure participant confidentiality.

## Protocol

This study was designed as a nonrandomized controlled trial in a laboratory setting. The design and protocol is shown in Fig 1.

The ICCs of joint angle (ROM) and force applied to an ankle during joint angle measurement between Pre-control and -second main trial (● and ▪) were analyzed to investigate the existence of the warm-up effect by control intervention.

The ICCs of joint angle (ROM) and force applied to the ankle during joint angle measurement at the Pre between the control and the second main trials (● and ▪, respectively) were analyzed to investigate the existence of the warm-up effect by control intervention. ICC, intraclass correlation; Lt, left; Post, immediately after intervention; Post10, 10 min after intervention; Pre, before intervention; ROM, range of motion; Rt, right.

Before the main test, four male and two female healthcare and conditioning specialists performed FR under free and maximum effort conditions thrice using a commercially available soccer ball (ERREJOTA, Adidas Co. Ltd., Germany; 1,000 hPa) for 20 s at 50 beats per min (bpm), or 25 strokes per min. Force (kgf) was monitored during FR with a force plate (9286BA, Kisler Co. Ltd., Switzerland). The average, maximum, and minimum forces

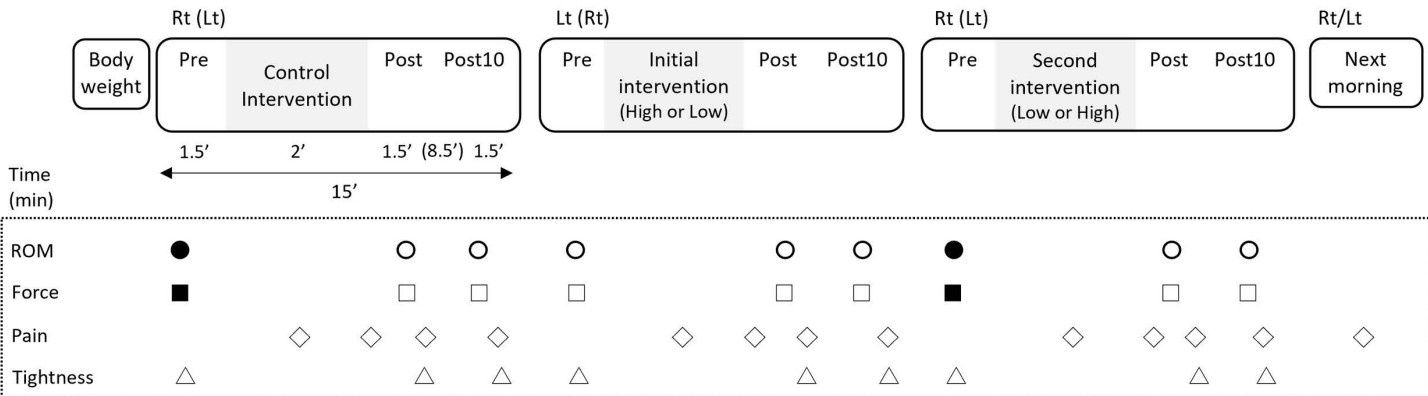

**Fig 1. An overview of the experimental protocol.**

per body weight (%) were: 34.6% (34.1–35.3%), 41.1 (40.3–42.6%), and 21.1% (18.4–24.7%), respectively, in the free condition, and 44.4% (44.3–45.0%), 50.7% (48.4–52.2%), and 39.1% (38.4–40.2%), respectively, in the maximum-effort condition. Based on these results, we adapted the minimum pressure during the free condition (15–25% of body weight) and maximum pressure during the maximum-effort condition (45–55% of body weight) for each low- and high-pressure conditions to confirm their differences. The force under high- and low-pressure conditions was equivalent to that in previous studies [23,24]. Before the experiment, regular calibration was conducted to ensure the reliability of devices used for ROM and strength measurements, including the force plate and hand dynamometer.

First, participants were interviewed regarding their age and current muscle soreness. Body weights were then measured to calculate the targeted pressure during FR, followed by the control intervention (CTRL) and FR under high (High) and low (Low) pressures for 2 min randomly (Fig 1). For CTRL, the participants placed their unilateral leg with a knee, kept it straight on a mobile board, and moved the leg back and forth for 2 min, maintaining a 50 bpm tempo (Fig 2a). For main intervention, participants placed their thighs on a soccer ball, set on the force plate, and rolled the ball back and forth for 2 min from the ischial tuberosity to the popliteal fossa, while maintaining a 50 bpm tempo (Fig 2b). The force (pressure) was measured by the force plate shown on the monitor so that a participant could confirm that the load applied to the ball met the target pressure (Fig 2c). The target leg was randomly selected. When the right leg was chosen as the CTRL, an initial intervention was performed with the left leg, followed by a second intervention with the right leg. The participants were asked to rest in the supine position immediately after each intervention (Post) until 10 min after the intervention (Post10). Two rounds of the passive straight leg raise (PSLR) test followed by the passive knee extension (PKE) test were conducted before the intervention (Pre) and at Post and Post10.

Three skilled examiners (licensed healthcare specialists) performed the PSLR and PKE tests. During both tests, the examiner measured the force applied to the posterior ankle just above the calcaneus by using a handheld dynamometer (MT-100; Sakai Co. Ltd., Japan) at the terminal position to confirm that the examiner measured the angle with an equivalent load. Another examiner visually confirmed that the non-measured leg was kept straight in a neutral

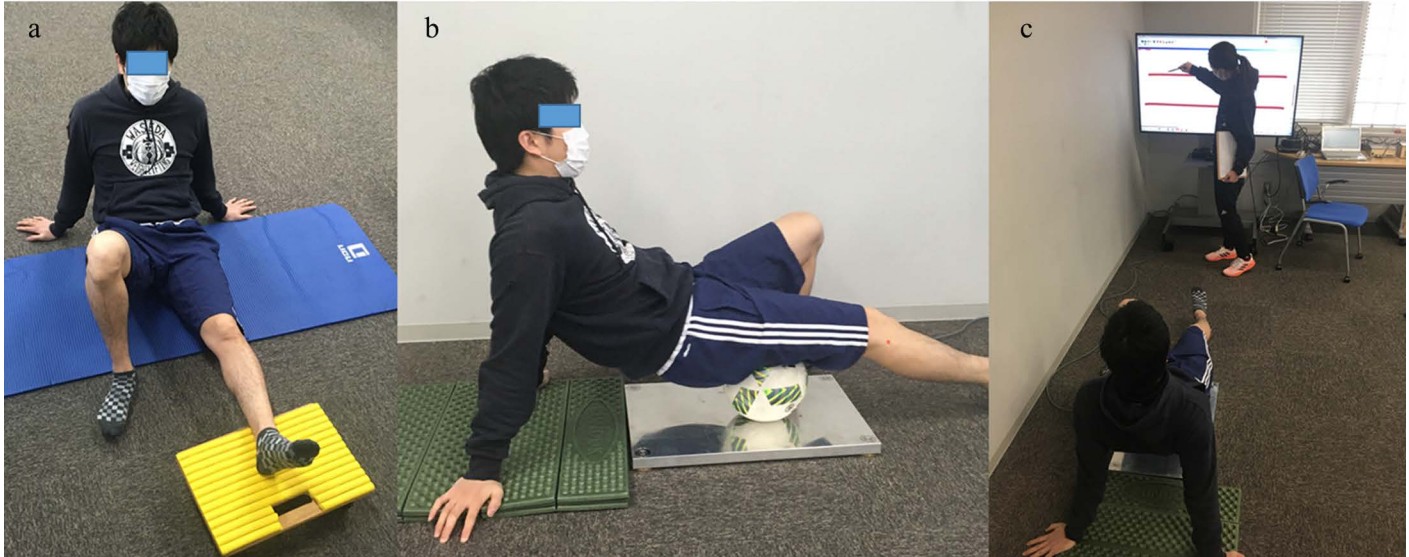

**Fig 2. An image of an experiment.** a (left): Control condition, b (center): Main trial, c (right): Monitoring force applied to the ball.

position, and the trunk and pelvis were kept in the initial position during both tests. Prior to the ROM measurement, the acromion, greater trochanter, lateral femoral epicondyle, fibular head, and lateral malleolus of both legs were marked.

In the PSLR test, participants laid in a supine position with their legs straight and the ankle of the tested leg relaxed. The examiner then kept the contralateral leg straight to avoid external or internal rotation and fixed the pelvis to minimize posterior pelvic tilt (maintained in the initial position). The examiner passively lifted the participant's legs during the hip flexion. The ankles were then fixed in a relaxed position. Per a previous study, PSLR endpoint was determined using at least one of these: (a) the examiner's perception of firm resistance and (b) palpable pelvic rotation [25]. Hip angle was measured using a universal goniometer. The examiner placed the axis of the goniometer on the line between the projection of the greater trochanter and the lateral epicondyle, with one arm placed parallel to the floor [26].

In the PKE test, the patient laid supine with the hip of the tested leg at 90° flexion. Using a goniometer, another skilled examiner confirmed that the hip angle was set at 90° flexion by measuring the angle between the acromion, greater trochanter, and lateral epicondyle. The examiner then extended the knee until they perceived firm resistance. The ankle was fixed in a relaxed position to minimize gastrocnemius muscles' influence. Subsequently, a skilled examiner measured knee angle using a universal goniometer. The goniometer axis was placed on the lateral epicondyle of the femur. The moving arm pointed toward the lateral malleolus and the stationary arm pointed toward the greater trochanter.

Intra-class correlations (ICCs) (1,1) between two rounds of measurement in ROM at each time point, ranging from 0.957–0.986 for PSLR and 0.900–0.969 for PKE, were considered "almost perfect" as in a previous study [27]. Thus, the averaged values of the two repetitions of each test were further analyzed.

The participants were interviewed about their perceived pain (hereafter referred to as pain) during each intervention and before and after the ROM test at Post and after Post10. (Fig 1) The muscle tightness participants felt pre- and during the ROM test was assessed immediately after the test. (Fig 1) The participants marked their pain and tightness on a visual analog scale, shown as a 100-mm bar, indicating no pain at 0 mm and the highest pain level at 100 mm. Pain was described as an arbitrary unit (AU). The perceived pain in the following morning was investigated on a 10-grade scale (0, no pain; 10, severe pain) via e-mail to confirm the participants did not get any injury or soreness.

## Statistical analyses

All data were presented as the average and standard error. ICCs of the ROM and force between the CTRL and main trials of the same limb (● in Fig 1) and among values at the Pre, Post, and Post10 time points during control trials were calculated to analyze CTRL influence on the main trial. A two-way mixed repeated-measures ANOVA was performed to analyze the differences in ROM, pain, and tightness during the tests at different time points and pressure levels and between different sexes. Additionally, a two-way ANOVA was performed to analyze pain differences between sexes and the three intervention types. The perceived pan in the following morning was investigated using a descriptive procedure. When a significant interaction or main effect was observed, Bonferroni's post hoc test was conducted. Effect sizes for the ANOVA and post hoc tests were assessed using partial $\eta^2$ and Cohen's d. Thresholds for partial $\eta^2$ were set as small (0.01), medium (0.06), and large (0.14) [28]. For Cohen's d, the thresholds were small (0.20), medium (0.50), and large (0.80) [28]. Statistical analyses were performed using SPSS ver. 25 (IBM Corp., Armonk, NY, USA). Statistical significance was defined as $p < 0.05$, except for $p < 0.001$ in the Results section. All tables indicate $p < 0.05$.

## Results

The average ROM, force, and ICCs of PSLR and PKE tests at the Pre in CTRL and main trials (● and ▪ in Fig 1) are shown in Table 1.

As shown in Table 2, significant interactions were observed for PSLR between time and pressure, and between time and sex. PSLR values at Post and Post10 were significantlygreater than Pre across all conditions (p < 0.001), except for males under the Low pressure condition, which also showed a significant increase (p = 0.006). In addition, effect size (d)

Effect sizes for comparisons among each condition are reported in Appendices 1 and 2. ranged from 0.67 to 1.16 (moderate to large effect) in females and 0.32 to 0.69 (small to moderate effect) in males. (Appendix 1) For PKE (Table 3), a significant interaction was found between time and pressure. Females showed significantly greater PKE values at Post and Post10 compared to Pre under both Low (p = 0.019 and p = 0.007, respectively) and High pressure conditions (p = 0.006 and p = 0.004, respectively). Similarly, males demonstrated higher PKE values at Post and Post10 compared to Pre for both Low (p < 0.001 and p = 0.001, respectively) and High pressure conditions (p = 0.022 and p = 0.002, respectively). The effect size ranged from 0.74 to 0.82 (moderate to large effect) in females and 0.49 to 0.65 (small to moderate effect) in males. (Appendix 1) Notably, no significant force interactions were observed for either PSLR or PKE (Tables 4 & 5).

Effect sizes for comparisons among each condition are reported in Appendices 3 and 4.Effect sizes for comparisons among each condition are reported in Appendices 3 and 4.

Table 6 shows no significant interaction between pressure and sex for pain levels during the interventions, and the post-hoc test revealed no significant differences. Nineteen out of

**Table 1. The ROM of PSLR and PKE and force applied to the ankle during measurement between pre-control and pre-main trial and ICCs of between data.**

|  |  | Pre-control trial | Pre-main trial | ICC(1,1) |
|---|---|---|---|---|
| PSLR | angle (°) | 83.9 (2.3) | 83.5 (2,4) | 0.958 |
|  | force (N) | 49.2 (2.7) | 47.8 (3.0) | 0.635 |
| PKE | angle (°) | 149.3 (2.2) | 147.7 (2.1) | 0.901 |
|  | force (N) | 39.9 (2.4) | 38.5 (2.3) | 0.904 |

PSLR: passive straight leg raise, PKE: passive knee extension. ICC: inter-class confidential interval

**Table 2. The ROM of PSLR at different time points.**

|  |  | Pre (°) | Post (°) | Post10 (°) |
|---|---|---|---|---|
| Female | CTRL | 86.0 (3.0) | 84.7 (3.1) | 85.2 (2.9) |
|  | Low | 84.1 (3.1) | 90.9 (3.3)* | 92.6 (2.5)* |
|  | High | 84.0 (2.9) | 95.4 (3.5)* | 94.8 (3.0)* |
| Male | CTRL | 81.9 (3.7) | 81.1 (3.9) | 80.7 (3.7) |
|  | Low | 81.1 (4.2) | 85.8 (3.8)* | 85.2 (3.8)* |
|  | High | 78.9 (3.8) | 85.3 (3.6)* | 87.1 (3.6)* |

*indicates p<0.05 vs. Pre. PSLR: passive straight leg raise. Statistical values are described below;

*Time x Pressure x Sex: F(4,72)=1.27, p=0.289, partial η2=0.05 (No significant)*

*Time x Pressure: F(4,72)=35.60, p<0.001, partial η2=0.57 (Large effect)*

*Time x Sex: F(2,36)=5.56, p=0.022, partial η2=0.09 (Moderate effect)*

*Pressure x Sex: F(2,36)=1.52, p=0.859, partial η2=0.01(No significant)*

Effect sizes for comparisons among each condition are reported in Appendices 1 and 2.

**Table 3. The ROM of PKE at different time points.**

| PKE | | Pre (°) | Post (°) | Post10 (°) |
|---|---|---|---|---|
| Female | CTRL | 152.1 (2.5) | 150.1 (2.4) | 151.9 (2.8) |
| | Low | 148.4 (2.7) | 154.1 (2.0)* | 155.8 (3.0)* |
| | High | 149.7 (2.7) | 156.4 (3.0)* | 156.8 (3.1)* |
| Male | CTRL | 146.6 (3.6) | 145.0 (3.4) | 144.1 (3.4) |
| | Low | 143.2 (4.1) | 150.8 (3.2)* | 149.5 (3.2)* |
| | High | 144.3 (3.4) | 148.7 (2.2)* | 150.3 (3.6)* |

*indicates p < 0.05 vs. Pre. PKE: passive knee extension. Statistical values are described below;

*Time x Pressure x Sex: $F_{(4,72)} = 0.92$, p = 0.912, partial $\eta^2 = 0.01$ (No significant)*

*Time x Pressure: $F_{(4,72)} = 15.34$, p < 0.001, partial $\eta^2 = 0.36$ (Large effect)*

*Time x Sex: $F_{(2,36)} = 1.22$, p = 0.274, partial $\eta^2 = 0.02$ (No significant)*

*Pressure x Sex: $F_{(2,36)} = 0.04$, p = 0.962, partial $\eta^2 = 0.01$ (No significant)*

**Table 4. The force applied to the posterior of the ankle during PSLR and ICCs among data at different time points.**

| PSLR | | Pre | Post | Post10 | ICC (1,1) |
|---|---|---|---|---|---|
| Female | CTRL | 45.0 (4.4) | 43.9 (3.5) | 43.9 (2.9) | 0.806 |
| | Low | 43.8 (4.5) | 48.0 (4.3) | 46.3 (3.9) | 0.880 |
| | High | 41.8 (3.5) | 46.3 (2.8) | 45.8 (2.7) | 0.810 |
| Male | CTRL | 53.5 (3.1) | 53.8 (3.0) | 56.4 (3.5) | 0.726 |
| | Low | 54.5 (3.9) | 56.4 (2.5) | 56.7 (3.3) | 0.747 |
| | High | 51.4 (3.7) | 56.8 (4.0) | 58.2 (2.0) | 0.656 |

PSLR: passive straight leg raise, ICC: inter-class confidential interval

Statistical values are described below;

*Time x Pressure x Sex: $F(4,72) = 1.81$, p = 0.174, partial η2 = 0.06 (No significant)*

*Time x Pressure: $F(4,72) = 0.91$, p = 0.125, partial η2 = 0.02 (No significant)*

*Time x Sex: $F(2,36) = 0.68$, p = 0.513, partial η2 = 0.04 (No significant)*

*Pressure x Sex: $F(2,36) = 0.07$, p = 0.934, partial η2 = 0.01 (No significant)*

**Table 5. The force applied to the posterior of the ankle during PKE and ICCs among data at different time points.**

| PKE | | Pre | Post | Post-10 | ICC (1,1) |
|---|---|---|---|---|---|
| Female | CTRL | 34.5 (2.8) | 32.6 (2.8) | 33.5 (3.0) | 0.616 |
| | Low | 32.8 (2.5) | 37.1 (3.5) | 41.0 (3.1) | 0.645 |
| | High | 34.8 (2.9) | 36.7 (2.6) | 36.7 (3.0) | 0.697 |
| Male | CTRL | 45.4 (3.3) | 46.1 (4.0) | 42.6 (2.5) | 0.741 |
| | Low | 44.5 (3.2) | 45.9 (3.1) | 44.7 (2.4) | 0.854 |
| | High | 45.0 (3.5) | 48.0 (2.5) | 46.5 (2.5) | 0.765 |

PKE: passive knee extension, ICC: inter-class confidential interval

Statistical values are described below;

*Time x Pressure x Sex: $F(4,72) = 2.13$, p = 0.085, partial η2 = 0.11 (No significant)*

*Time x Pressure: $F(4,72) = 2.13$, p = 0.085, partial η2 = 0.11 (No significant)*

*Time x Sex: $F(2,36) = 2.53$, p = 0.094, partial η2 = 0.12 (No significant)*

*Pressure x Sex: $F(2,36) = 1.31$, p = 0.282, partial η2 = 0.07 (No significant)*

**Table 6. Visual analog scale (AU) of pain during FR interventions.**

|  |  | Pain |
|---|---|---|
| Female | CTRL | 5.4 (2.9) |
|  | Low | 4.8 (2.6) |
|  | High | 17.1 (7.4) |
| Male | CTRL | 4.8 (3.4) |
|  | Low | 5.7 (1.9) |
|  | High | 13.4 (4.5) |

$F_{(2,36)} = 0.16$, p = 0.854, partial $\eta^2 = 0.01$

Effect sizes for comparisons among each condition are reported in Appendix 5.

20 participants reported a pain level of zero in the following morning. One male participant reported pain levels of three on the left leg (Low) and four on the right leg (CTRL and High).

Table 7 shows no significant interactions between time, pressure, and sex in pain felt during the ROM tests. Conversely, a significant interaction was observed in tightness during the ROM tests between time and pressure, with no difference in the post-hoc test. (Table 8)

## Discussion

This study aimed to investigate how the magnitude of applied pressure and sex influenced acute ROM changes immediately after FR intervention. We found that acute hip- (PSLR) and knee- (PKE) joint ROM increased with FR intervention of the hamstring muscles. Furthermore, no interaction exists between pressure and sex, suggesting that applying pressure to the FR hamstring muscles may effectively induce an acute ROM increase at the knee or hip joints, regardless of sex.

Consistent with our hypothesis, pressure and rolling stimuli applied to the hamstring muscles using commercially available soccer balls improved the acute ROM in PSLR and PKE without any increased feeling of tightness or perceived pain. The acute ROM changes observed in this study were comparable with those reported in previous studies using specialized equipment. This finding highlights the potential of using readily available tools, such as a soccer ball, to achieve similar mobility benefits. Such an approach could be particularly beneficial for athletes and physical therapy patients who may not have access to specialized FR equipment. FR intervention in the hamstring muscles using FR equipment reportedly improved the ROM of PSLR by 2.58% (from 123.23 ± 3.9° to 126.41 ± 3.62°) in young and recreationally active females [29]. In female participants, acute ROM improvement of PSLR was 8.21% (from 84.1 ± 3.1° to 90.0 ± 3.3°) under low pressure and manu13.50% (from 83.6 ± 2.8° to 95.4 ± 3.5°) under high pressure. In a previous study, while no significant acute improvement was observed in chair-seated passive knee extension at 5 min after FR intervention (approximately 1.5%) [17], male participants of this study demonstrated substantial acute improvement of PKE in 5.4% (from 143.2 ± 4.1° to 150.9 ± 3.2°) under low pressure, and 3.9% (144.8 ± 3.4° to 150.5 ± 3.6°) under high pressure 10 min following intervention. These findings indicate that even commercially available soccer balls can temporarily improve knee extension and hip flexion ROM by applying pressure and rolling stimuli to the hamstring muscles. Hence, applying mechanical stimuli, such as pressure and rolling, is crucial for obtaining temporal ROM improvement in FR interventions, regardless of equipment type. Manual massage, which involves physical pressure stimulation of the skin, has been shown to enhance muscle extensibility, which may contribute to changes in ROM [30]. Similarly, roller exercises applying pressure and rolling stimuli demonstrated ROM improvements comparable with FR [31],

**Table 7. Visual analog scale (AU) of pain during PSLR and PKE.**

| Pain | | Pre-ROM measurement | Post | Post10 |
|---|---|---|---|---|
| Female | CTRL | 19.8 (6.2) | 13.3 (4.7) | 10.0 (3.8) |
| | Low | 14.9 (5.2) | 12.4 (3.3) | 10.4 (3.7) |
| | High | 14.3 (4.2) | 13.7 (4.4) | 17.5 (4.8) |
| Male | CTRL | 14.8 (9.1) | 11.5 (7.7) | 16.1 (8.4) |
| | Low | 23.4 (9.8) | 20.0 (9.0) | 20.7 (9.5) |
| | High | 13.2 (7.7) | 11.0 (5.6) | 11.8 (5.7) |

PSLR: passive straight leg raise, PKE: passive knee extension

Statistical values are described below;

Time x Pressure x Sex: $F_{(4,72)} = 1.81$, $p = 0.174$, partial $\eta2 = 0.06$ (No significant)

Time x Pressures: $F_{(4,72)} = 0.91$, $p = 0.408$, partial $\eta2 = 0.03$ (No significant)

Time x Sex: $F_{(2,36)} = 0.67$, $p = 0.418$, partial $\eta2 = 0.01$ (No significant)

Pressure x Sex: $F_{(2,36)} = 0.50$, $p = 0.611$, partial $\eta2 = 0.02$ (No significant)

Effect sizes for comparisons among each condition are reported in Appendices 6 and 7.

**Table 8. Visual analog scale (AU) of tightness during PSLR and PKE.**

| Tightness | | Pre | Post | Post10 |
|---|---|---|---|---|
| Female | CTRL | 35.8 (7.8) | 35.7 (6.6) | 31.0 (6.1) |
| | Low | 37.6(7.7) | 41.1 (5.8) | 38.5 (6.4) |
| | High | 30.7 (5.4) | 43.5 (6.7) | 44.8 (6.4) |
| Male | CTRL | 43.5 (9.7) | 45.7 (7.4) | 49.7 (7.3) |
| | Low | 44.0 (8.2) | 54.9 (6.8) | 52.2 (8.0) |
| | High | 38.0 (7.1) | 46.5 (6.5) | 47.0 (6.5) |

PSLR: passive straight leg raise, PKE: passive knee extension

Statistical values are described below;

Time x Pressure x Sex: $F_{(4,72)} = 0.77$, $p = 0.393$, partial $\eta2 = 0.04$ (No significant)

Time x Pressure: $F_{(4,72)} = 6.43$, $p = 0.021$, partial $\eta2 = 0.26$ (Large effect)

Time x Sex: $F_{(2,36)} = 0.70$, $p = 0.413$, partial $\eta2 = 0.04$ (No significant)

Pressure x Sex: $F_{(2,36)} = 0.30$, $p = 0.591$, partial $\eta2 = 0.02$ (No significant)

Effect sizes for comparisons among each condition are reported in Appendices 8 and 9.

indicating that applying pressure or pressure plus rolling stimulation to the muscle or soft tissue around the muscle effectively enhances acute ROM, regardless of the equipment used. This finding seems reasonable, based on the possible mechanism of immediate ROM changes with FR intervention. FR-induced thixotropic effects [32], such as an increase in blood flow after FR [14], contribute to an acute ROM increment. The rolling on the skin, fascia, and muscle generates friction, which may increase tissue temperature and potentially enhance range of motion (ROM). Additionally, rolling may apply shear stress to tissues, reducing intra- and inter-cellular fluid viscosity and making movement less resistant. It is plausible that these mechanical effects, including friction and shear stress, contribute to temporary changes in ROM. These findings underscore the potential of physical stimuli, such as pressure and rolling, to influence ROM [33]. FR interventions may acutely improve ROM through mechanisms such as muscle relaxation and possible changes in sympathetic nerve activity [11,12], though further research is needed to confirm these pathways.

This study also revealed a significant interaction between time and pressure for PSLR and PKE. However, no significant difference was observed between low and high pressures, consistent with a previous study that showed no intensity-dependent difference in ROM change after FR in the quadriceps muscles [5]. The hamstring muscles may respond more noticeably to FR than the calf muscles, while quadriceps effects were inconclusive due to data variability [1], and pressure magnitude has a limited impact on acute ROM changes. ROM changes were observed after FR at low intensity, whereas perceived pain during FR at low intensity was equivalent to that in the control condition. Participants indicated a perceived pain level of 13.7–17.1 AU per 100 AU, even during FR under high pressure, suggesting that FR intervention's high pressure and perceived pain were unnecessary to alter ROM. Perceived pain threshold modulation is influenced by DNIC, where remote perceived pain stimuli can impact central pain thresholds [18,19]. Research involving thermal stimulation of the hand and forearm to induce demonstrated that highly intense, subjectively non-painful stimulation can evoke pain-inhibitory effects. Subjectively non-painful stimuli are susceptible to inhibitory influences, suggesting that non-painful stimulation can elicit pain-inhibitory effects and subjective non-painful stimuli are influenced by inhibitory mechanisms during heterotopic noxious conditioning stimulation [21]. Furthermore, Granot et al. (2008) explored endogenous analgesic responses by varying water temperatures (12, 15, and 18°C for cold; and 44 and 46.5°C for warm) and skin temperature [33°C]) [22], revealing that analgesic effects were observed only at 12 and 46.5°C. Researchers found no significant correlation between the subjective perceived pain score and analgesia level, suggesting that physical stimuli application is independent of the subjective perceived pain level. Aboodarda et al. [9] reported that the pain level during FR does not necessarily affect pain threshold change [9], aligning with our findings. If stimuli intensity, such as pressure and rolling, has less influence on DNIC level, further research is warranted to investigate the minimum pressure and rolling of the FR required to elicit DNIC and increase acute ROM changes.

In contrast to our hypothesis, the sex difference in FR-induced acute improvement in ROM was not statistically significant. While female participants showed a larger effect size compared to males, the results do not sufficiently support the suggestion of a previous study that female participants are more sensitive to FR interventions [1]. While our results do not clarify why there were no sex differences in FR-induced ROM changes, we believe this is an opportunity to reconsider sex differences in FR-induced acute ROM change [1], considering sex differences in the physiological mechanisms of FR intervention-induced ROM changes. Regarding DNIC, several studies have indicated that males demonstrate greater changes in pain thresholds in response to noxious stimuli than females, highlighting a significant sex difference in this physiological response [22,34]. Arendt-Nielsen et al. [34] reported that cold pressor pain increased the pain threshold in both males and females, with greater increases observed in males. However, hypertonic saline-evoked muscle pain significantly increased the pain threshold in males but not in females [34], indicating that nociceptive stimuli, typically of a mechanical nature, may have a greater tendency to increase pain thresholds in males than in females. However, the extent to which this occurs may depend on the stimulus type. Thus, further research is needed regarding sex differences in FR-induced acute ROM changes by adopting various FR conditions such as speed, duration, and surface of the equipment to modulate stimulation. Furthermore, the above findings imply the need for more individualized conditions to elicit acute ROM changes due to FR.

This study has some methodological limitations. First, two different FR pressures were applied to the contralateral side on the same day. Recent studies have implied the existence of a crossover effect in FR [35,36] However, the ROM values before the interventions did not differ between the low- and high-pressure conditions. Nevertheless, our results may have been

influenced by this intervention design. Future studies could address this limitation by separating the pressure conditions across different days to minimize potential crossover effects. Second, the applied force was monitored using a handheld dynamometer. Moderate-to-good ICC may have also limited our results to PKE [27]. To improve measurement precision, future studies could utilize fixed setups for handheld dynamometers or alternative devices such as digital torque meters or load cells. Lastly, the participants in this study were well-trained collegiate athletes with no history of injury to the target muscles. Thus, our findings may be limited to well-trained young adults without specific muscular conditions, such as induration. Future research should include more diverse participant groups, such as individuals with varying fitness levels, older adults, or those with muscular conditions, to increase the generalizability of the findings.

Despite these limitations, our study provides valuable insights into the effects of FR intensity and sex on acute ROM changes. The use of a commercially available soccer ball as an FR equipment highlights the potential for practical, cost-effective interventions that do not require specialized equipment. This approach could be particularly beneficial for athletes and physical therapy patients who may lack access to conventional FR equipment. Future studies should explore the long-term effects of FR using different equipment, investigate additional variables such as vibration and rolling speed, and assess its impact on populations with specific needs, such as individuals with limited mobility or chronic pain conditions.

## Conclusion

Applied pressure and rolling stimuli on targeted muscles are effective methods for improving acute ROM changes, regardless of sex. While FR pressure magnitude and the degree of perceived pain may not strongly affect acute ROM changes, the effective conditions for inducing such changes appear to have greater individual specificity than previously recognized. Therefore, further research is needed to identify individual effective conditions, particularly by exploring diverse attributes of stimuli, such as rolling speed, duration, and the surface texture of the equipment. Additionally, investigations into sex differences in FR-induced acute ROM changes under varying FR conditions will provide valuable insights into personalized intervention strategies.

## Supporting information

**Appendix 1. Effect sizes of time-point differences in ROM of PSLR and PKE by sex and intensity levels.**
(DOCX)

**Appendix 2. Effect sizes of intensity level comparisons in ROM of PSLR and PKE across time points by sex.**
(DOCX)

**Appendix 3. Effect sizes of applied force differences across time points in PSLR and PKE by sex and intensity.**
(DOCX)

**Appendix 4. Effect sizes of applied force comparisons across intensity levels in PSLR and PKE by sex and time points.**
(DOCX)

**Appendix 5. Effect size of pain comparisons across intensity levels during FR intervention by sex.**
(DOCX)

**Appendix 6. Effect size of pain comparisons across time points during ROM measurements by sex and intensity levels.**
(DOCX)

**Appendix 7. Effect size of pain comparisons across intensity levels during ROM measurements by sex and time points.**
(DOCX)

**Appendix 8. Effect size of tightness comparisons across time points during ROM measurements by sex and intensity levels.**
(DOCX)

**Appendix 9. Effect size of tightness comparisons across intensity levels during ROM measurements by sex and time points.**
(DOCX)

## Acknowledgments

The authors appreciate Dr. Daichi Nishiumi's kind assistance for recruiting participants.

## Author contributions

**Conceptualization:** Norikazu Hirose.

**Data curation:** Norikazu Hirose.

**Formal analysis:** Norikazu Hirose.

**Investigation:** Norikazu Hirose, Akane Yoshimura, Kei Akiyama, Atsuya Furusho.

**Methodology:** Norikazu Hirose, Akane Yoshimura, Kei Akiyama.

**Project administration:** Norikazu Hirose.

**Resources:** Norikazu Hirose, Akane Yoshimura, Kei Akiyama.

**Supervision:** Norikazu Hirose.

**Validation:** Norikazu Hirose.

**Visualization:** Norikazu Hirose.

**Writing – original draft:** Norikazu Hirose.

**Writing – review & editing:** Norikazu Hirose, Akane Yoshimura, Kei Akiyama, Atsuya Furusho.

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
