## [Decision Letter · Decision Letter 0]

26 Nov 2024

PONE-D-24-45387A sex and pressure-dependent difference in acute range of motion after foam rolling on hamstring musclesPLOS ONE

Dear Dr. Hirose,

 Thank you for submitting your manuscript to PLOS ONE. After careful consideration, we feel that it has merit but does not fully meet PLOS ONE’s publication criteria as it currently stands. Therefore, we invite you to submit a revised version of the manuscript that addresses the points raised during the review process.

**ACADEMIC EDITOR: ** your manuscript has been reviewed by two experts in the filed that reported several minor issues you should consider while revising your work

We look forward to receiving your revised manuscript.

Kind regards,

Emiliano Cè, Ph.D.

Academic Editor

PLOS ONE

Journal Requirements:

Reviewers' comments:

Reviewer's Responses to Questions

**Comments to the Author**

1. Is the manuscript technically sound, and do the data support the conclusions?

Reviewer #1: Yes

Reviewer #2: Yes

2. Has the statistical analysis been performed appropriately and rigorously? 

Reviewer #1: Yes

Reviewer #2: Yes

3. Have the authors made all data underlying the findings in their manuscript fully available?

Reviewer #1: Yes

Reviewer #2: Yes

4. Is the manuscript presented in an intelligible fashion and written in standard English?

Reviewer #1: Yes

Reviewer #2: Yes

5. Review Comments to the Author

Reviewer #1: Dear Authors,

Your study is a rigorous investigation into the effects of foam rolling on acute range of motion (ROM) in the fields of sports science and physical therapy. The article presents a comprehensive methodology and detailed statistical analyses. However, with certain refinements, I believe your work could become more accessible to a broader readership and align more closely with PLOS ONE standards.

Title and Abstract: Restructuring the title and abstract to present the main findings in a more straightforward manner could facilitate quick comprehension for readers. In particular, reducing technical details in the abstract to highlight general findings and practical applications would better convey the significance of the study.

Ethics and Data Access: A more detailed ethical statement in accordance with the Declaration of Helsinki and a clarification of data accessibility are recommended. Such additions will enhance the study's credibility and allow readers the option to verify findings.

Presentation of Results: Simplifying tables and graphics and focusing on key findings may improve the readability of the results. Emphasizing primary findings through visuals will help readers grasp the study’s implications more efficiently.

Future Research and Limitations: It is suggested to expand the discussion to include the study’s potential applications and future research directions. Clearly addressing limitations and providing concrete suggestions for future studies would enhance the study’s academic value.

I believe that revisions based on these suggestions will strengthen your manuscript and broaden its appeal to a wide audience.

Sincerely,

Reviewer #2: Thank you for the opportunity to review this article. This paper seems really a good contribution to the literature. I congratulate authors for their efforts and well-written methodology and results parts in particular. Firstly, can you check the link for your data? I could not open it. I have some other concerns that I have explained below in detail. Moreover, for some citations throughout the paper, I suggest authors to check them, and in some cases, please go to the very first paper saying/showing/presenting/giving that specific finding or suggestions. Also, I believe that a more critical perspective will add value for the readers considering different moderators in FR such as gender.

Congratulations again!

Please see my specific comments below.

Line 42: Can you check the citations here. You have mentioned here that FR increases ROM but I do have some concerns regarding the references you have here. I just checked some of them and saw some confusion. In citation 1, there are findings favoring foam rolling group in jump, agility, bench press and sprint, so, not only in sprint. Furthermore, I suggest you to consider Mikesky’s study cautiously because the focus is not ROM in their study, there is no performance improvement in any parameter, and also they do not have much data regarding ROM to discuss. It is similar to reference number 3. There is no performance difference, yet there are some parameters such as fatigue. Also, ROM is not a focus in this study either. You should also consider the results in reference number 5. There are different vertical jump findings in SMR_5 and SMR_1 conditions. You have already mentioned “1-2 min interventions” in your paper but may be adding one or two sentence to explain time-related issues while applying FR for different performance parameters might inform the readers better in introduction part. Overall, can you please reconsider and confirm whether these references suggest FR increases ROM cause this is what you are saying in the core of this sentence.

Line 48-51: Needs rewriting. I guess you are trying to say that the effects observed in studies including only men are smaller than the ones including females or mixed samples. Then, say it simply. You can also divide the sentence into two in order to make it easier to follow and understand for all.

Line 77-78: Needs rewriting.

Line 100-101: Can you divide the sentence into two to make it easier to follow as “… athletes training for at least 3 days a week . Our exclusion criteria were: current injury ….” ?

Line 103: Can you change it to: A total of 20 athletes (10 female and 10 male) participated …

Line 105: Can you change it to: They were asked not to engage in exhausting activity….

Line 109: Approval number: 2020-418

For participants, G*Power analysis suggested 40 participants but you only included 20?

The authors explained the study protocol in detail.

Line 184: … ”almost perfect” as in a previous …

In the whole text, for the word “pain” that you have evaluated, it is better to use “perceived pain” because you just ask participants regarding their thoughts and perceptions. So, can you change them into “perceived pain”?

Line 202: The perceived pain in the following morning was investigated…

Line 204: No need for that sentence: “Statistical significance was set at p<0.05.” Because you also explain it in the next lines.

Line 226: .. in the following morning.

All tables were prepared good and explained well in the comments in results section.

Line 243: … hamstring muscles using commercially available soccer balls improved…

Line 250-254: Can you please check the article you have cited here (number 21)? Can you check the numbers and the other details?

Line 258: Instead of device, use tool or equipment. (You have used “equipment” in the following sections, so changing it to “equipment” might work better, but you can decide. Just make sure that it is coherent with the following statements as well.)

Line 260: Check the cited articles (36-38) whether they really say so.

Line 266: Use the citation below instead of both studies (13,9) because they also cited this article while discussing about higher blood flow. They do not have any data showing the case for higher blood flow but Hotfiel et al does! I suggest you to go to the original study. I think it is more ethical to give the credits to those who conduct the original study instead of citing others.

Hotfiel T, Swoboda B, Krinner S, Grim C, Engelhardt M, Uder M, et al. Acute effects of lateral thigh foam rolling on arterial tissue perfusion determined by spectral doppler and power doppler ultrasound. J Strength Cond Res. 2017;31(4):893–900.

Line 266-270: Needs rewriting.

Line 270-281: I do not think you need to give that much place for such discussions because it is not a part of your study, you do not have any sympathetic activity data. Maybe, you can just mention it in a sentence and go through your main focus and data-related discussions.

Line 288: Can you check your statement here and your citation. Can you confirm Wilke et al has any related part in their article saying hamstring is more sensitive to FR than quadriceps? On the contrary, there is a part in Wilke et al saying:

“The rolled muscle seems to represent a second relevant moderator. The strongest effect was seen in the hamstrings, whose treatment leads to higher increases than FR of the calf muscles (significant difference between levels, p = 0.008). Despite a large effect size, treatment of the quadriceps failed statistical significance (p > 0.05) which is, however, ascribed to the large data variability and the broad confidence interval.”.

So, please think about this statement again.

Line 305: Correct the citation as Aboodarda et al. [14]… .

Line 319: Correct the citation as Arendt-Nielsen et al. [number]…

Line 342: “Vital” is not a good word choice. They are just two options along with some other ways. So, you can only say your findings suggested that applying pressure and rolling stimuli on targeted muscles are good ways to improve acute ROM changes … .

In conclusion part, you can add similar sentence to the sentence you have in line 325-326 (further research is needed regarding sex differences in FR-induced acute ROM changes by adopting various FR conditions such as speed, duration, and surface of the equipment to modulate stimulation) to highlight the need for considering different moderators.

Overall comment: I congratulate authors for their work. The manuscript seems strong but I have some concerns with some parts that I have explained above. So, I suggest minor revision and would like to see the paper again after revisions if the authors accept to revise it.

6. PLOS authors have the option to publish the peer review history of their article (what does this mean? ). If published, this will include your full peer review and any attached files.

**Do you want your identity to be public for this peer review?** For information about this choice, including consent withdrawal, please see our Privacy Policy .

Reviewer #1: No

Reviewer #2: No

---

## [Author Response · Author response to Decision Letter 1]

12 Dec 2024

Reviewer #0

Thank you very much for your thorough and thoughtful review of our manuscript. We sincerely appreciate the time and effort you have invested in providing detailed and constructive feedback. Your insightful comments have been invaluable in improving the quality and clarity of our work. We have carefully addressed all the points raised in your review and made the necessary revisions to the manuscript. A detailed response to each comment has been included in the letter for your reference. We hope that the revised manuscript meets your expectations and aligns with the high standards of the journal.

Title and Abstract

Title: While the title reflects the content well, it includes technical language and could be simplified. A suggested revision might make the title more accessible to a broader readership.

Suggestion: A clearer title, such as “Gender and Pressure Effects of Foam Rolling on Acute Range of Motion in the Hamstring Muscles,” could be more understandable.

Response: Thank you for a valuable suggestion. We carefully considered the reviewer’s suggestion and modified the title to “Sex and Pressure Effects of Foam Rolling on Acute Range of Motion in the Hamstring Muscles"

Abstract: According to PLOS ONE’s requirements, the abstract should be brief, clear, and highlight the main results. The current abstract contains excessive technical details, which may make it difficult for readers to grasp the key findings.

Suggestion: The abstract should be restructured to focus on the primary findings and significance of the study. For instance, numerical details like specific ROM values may overshadow the main findings and could be omitted.

Response: Thank you for a valuable suggestion. We revised the abstract to avoid redundant explanations about technical details.

Introduction

Background Information: The introduction provides comprehensive information on the importance of ROM and foam rolling. However, some sentences are overly lengthy and complex, which hinders readability.

Suggestion: Simplify sentences such as "Foam rolling is a beneficial technique that increases ROM immediately after intervention without a reduced performance of sprint, jump, and strength" to improve clarity, e.g., “Foam rolling effectively increases range of motion without decreasing sprint, jump, or strength performance.”

Response: Thank you for your constructive feedback. We have revised the section to simplify overly complex sentences and improve clarity while retaining all essential information. We believe this adjustment enhances readability and aligns the content more closely with the reader’s expectations.

Hypothesis and Aim: The study’s aim and hypothesis are clearly defined but could be emphasized further.

Suggestion: A more explicit statement such as, “The aim of this study is to investigate the acute effects of foam rolling on ROM based on gender and pressure” would enhance clarity.

Response: Thank you for your suggestion. We have revised the statement of the study’s aim and hypothesis to enhance clarity and emphasis, as suggested. The updated text provides a more explicit and straightforward description of the study’s purpose and hypothesis, aligning with the feedback to improve readability and focus. (L74-79)

Materials and Methods

Ethics Statement and Participant Consent: According to PLOS ONE standards, studies involving human participants must clearly indicate ethics board approval and participant consent. While the ethics approval number is provided, further detail on the approval process is lacking. Per PLOS ONE guidelines, the study should state that it was conducted in accordance with the Declaration of Helsinki and that written consent was obtained from all participants.

Suggestion: Include a statement such as, “This study was conducted in accordance with the Declaration of Helsinki and approved by the Ethics Committee of XXXXX University (approval number: 2020-418). Written consent was obtained from all participants.”

Response: Thank you very much for your valuable comment. In accordance with all the reviewers’ comments, we revised more detail as follows.

L102-109: This study was conducted in full compliance with the Declaration of Helsinki. Ethical approval was obtained from the Ethics Committee of Waseda University, approval number 2020-418. All participants were provided comprehensive information regarding the study objectives, procedures, potential risks, and benefits. Written informed consent was obtained from each participant prior to their involvement in the study. Participants were informed that their participation was voluntary and that they could withdraw from the study at any time without any repercussions. Furthermore, all data collected were anonymized to ensure participant confidentiality.

Data Accessibility: PLOS ONE requires clear accessibility of research findings and data, especially for reuse purposes. It is stated that the data are stored in a third-party repository, but the access details and DOI are unclear.

Suggestion: Add a statement like, “All data supporting this study’s findings are publicly available in the Dryad Digital Repository (DOI: https://doi.org/10.5061/dryad.1ns1rn93x).” (Minor Revision)

Response: Thank you for your comment. We revised it following the reviewer’s comment. The submitted data will be available after the curation process by Dryad.

Measurement Methods: While the methods for measuring ROM and strength are described in detail, there is a lack of information regarding the calibration and reliability of the equipment used. No details are provided about the calibration of devices such as the force plate and hand dynamometer.

Suggestion: Add a statement like, “Regular calibration was conducted to ensure the reliability of devices used for ROM and strength measurements.” (Minor Revision)

Response: Thank you for your valuable comment. We agree that including information about the calibration and reliability of the equipment is essential for transparency and rigor. We have revised the Methods section to include the following statement:

L133-135: Before the experiment, regular calibration was conducted to ensure the reliability of devices used for ROM and strength measurements, including the force plate and hand dynamometer.

Statistical Analyses: Although the statistical analyses are generally well-explained, PLOS ONE guidelines recommend reporting effect sizes alongside significance tests. ANOVA and post-hoc tests were used, but effect sizes (e.g., partial eta-squared) were not reported.

Suggestion: Report effect sizes, such as η² or Cohen’s d, alongside significance values in the statistical analysis. (Minor Revision)

Response: Thank you for your insightful comment. We have revised the Statistical Analyses section to include effect sizes (e.g., Cohen’s d) alongside significance tests. For post-hoc comparisons, effect sizes are reported for all pairs where significant differences were observed after Bonferroni correction. Additionally, for completeness, all effect sizes, including those for non-significant comparisons, are presented in the Appendix to provide a comprehensive overview of the results. In the main text, only the range of effect sizes for post-hoc comparisons with statistically significant differences is reported. Moreover, partial η² values for the ANOVA interactions are now included in the table footnotes to provide further context regarding the magnitude of effects observed in the study. We also acknowledge that the use of partial η² for ANOVA was not initially mentioned in the Methods section. This has now been rectified, ensuring consistency between the Methods and Results sections.

L203-208: Effect sizes for the ANOVA and post hoc tests were assessed using partial η² and Cohen’s d. Thresholds for partial η² were set as small (0.01), medium (0.06), and large (0.14)[28]. For Cohen’s d, the thresholds were small (0.20), medium (0.50), and large (0.80)[28].

In response, we have revised the relevant section to incorporate the finding that female participants demonstrated a larger effect size compared to males, while emphasizing that no statistically significant difference was observed. The revised sentence is as follows:

L313-319: In contrast to our hypothesis, the sex difference in FR-induced acute improvement in ROM was not statistically significant. While female participants showed a larger effect size compared to males, the results do not sufficiently support the suggestion of a previous study that female participants are more sensitive to FR interventions

Results

Tables and Figures: The results section includes numerous tables and detailed data, but some tables may hinder readers from understanding the findings clearly.

Suggestion: Simplify tables to focus on key findings. For instance, general trends by gender and pressure could be summarized, focusing on tables that present significant statistical differences. (Minor Revision) Clarity of Statistical Findings: Excessive numerical details in the results may obscure the main findings.

Suggestion: Emphasize the main findings (e.g., differences in ROM by gender and pressure) and avoid excessive detail on minor subgroup analyses. (Minor Revision)

Response: Thank you for your comments. To improve clarity and focus on the main findings, we revised the results section, including tables, by simplifying the description and emphasizing the significant interactions and key differences. The revised text maintains the statistical rigor while making the findings more accessible to readers.

Discussion

Interpretation and Significance of Findings: The discussion addresses the study’s findings thoroughly, but the broader implications and practical applications are not clearly conveyed.

Example Sentence: "The acute changes in ROM observed in this study were comparable with those reported in previous studies.”

Suggestion: To provide broader context, consider a statement like, “The acute ROM changes observed in this study have potential benefits for enhancing mobility in athletes and physical therapy patients.” (Minor Revision)

Response: Thank you for your insightful comment. We have revised the discussion to better convey the broader implications and practical applications of our findings. The updated text emphasizes that the use of readily available tools, such as a soccer ball, can achieve similar ROM benefits to those obtained with specialized equipment. This revision highlights the potential applicability of our findings to athletes and physical therapy patients who may lack access to specialized devices.

L250-252: The acute ROM changes observed in this study were comparable with those reported in previous studies using specialized equipment. This finding highlights the potential of using readily available tools, such as a soccer ball, to achieve similar mobility benefits. Such an approach could be particularly beneficial for athletes and physical therapy patients who may not have access to specialized FR equipment.

Limitations and Future Research: The limitations of the study (e.g., the use of two different pressure applications in a single day) are well-explained, but more specific suggestions for future research could be provided.

Suggestion: Add concrete suggestions, such as, “Future studies could explore the effects of different foam rolling devices or include larger and more diverse participant groups.” (Minor Revision)

Response: Thank you for your thoughtful comment. We have revised the Limitations section to include more concrete suggestions for future research, such as separating pressure conditions across different days, employing more precise monitoring systems, and including more diverse participant groups. These additions aim to address the study's limitations and provide a clearer direction for future studies. (L333-348)

Reviewer #1:

Thank you very much for your thorough and thoughtful review of our manuscript. We sincerely appreciate the time and effort you have invested in providing detailed and constructive feedback. Your insightful comments have been invaluable in improving the quality and clarity of our work. We have carefully addressed all the points raised in your review and made the necessary revisions to the manuscript. A detailed response to each comment has been included in the letter for your reference. We hope that the revised manuscript meets your expectations and aligns with the high standards of the journal.

Title and Abstract: Restructuring the title and abstract to present the main findings in a more straightforward manner could facilitate quick comprehension for readers. In particular, reducing technical details in the abstract to highlight general findings and practical applications would better convey the significance of the study.

Response: Thank you for a valuable suggestion. We modified the title to “Sex and Pressure Effects of Foam Rolling on Acute Range of Motion in the Hamstring Muscles” in accordance with another reviewer's suggestion. We also edited the abstract to avoid redundant explanations about technical details.

Ethics and Data Access: A more detailed ethical statement in accordance with the Declaration of Helsinki and a clarification of data accessibility are recommended. Such additions will enhance the study's credibility and allow readers the option to verify findings.

Response: Thank you for your suggestion. We explain a more detailed ethical statement as follows:

L102-109: This study was conducted in full compliance with the Declaration of Helsinki. Ethical approval was obtained from the Ethics Committee of Waseda University, approval number 2020-418. All participants were provided comprehensive information regarding the study objectives, procedures, potential risks, and benefits. Written informed consent was obtained from each participant prior to their involvement in the study. Participants were informed that their participation was voluntary and that they could withdraw from the study at any time without any repercussions. Furthermore, all data collected were anonymized to ensure participant confidentiality.

Presentation of Results: Simplifying tables and graphics and focusing on key findings may improve the readability of the results. Emphasizing primary findings through visuals will help readers grasp the study’s implications more efficiently.

Response: Thank you for your comments. To improve clarity and focus on the main findings, we revised the results section by simplifying the description and emphasizing the significant interactions and key differences. The revised text maintains the statistical rigor while making the findings more accessible to readers.

Future Research and Limitations: It is suggested to expand the discussion to include the study’s potential applications and future research directions. Clearly addressing limitations and providing concrete suggestions for future studies would enhance the study’s academic value.

Response: Thank you for your thoughtful comment. We have expanded the discussion to include the study’s potential applications and future research directions. The revised section provides a clearer outline of the study's limitations and includes concrete suggestions for addressing them in future studies. Additionally, we have highlighted the practical implications of our findings to enhance the study's academic and practical value. (L313-332)

Reviewer #2:

Thank you very much for your thorough and thoughtful review of our manuscript. We sincerely appreciate the time and effort you have invested in providing detailed and constructive feedback. Your insightful comments have been invaluable in improving the quality and clarity of our work. We have carefully addressed all the points raised in your review and made the necessary revisions to the manuscript. A detailed response to each comment has been included in the letter for your reference. We hope that the revised manuscript meets your expectations and aligns with the high standards of the journal.

Line 42: Can you check the citations here. You have mentioned here that FR increases ROM but I do have some concerns regarding the references you have here. I just checked some of them and saw some

---

## [Decision Letter · Decision Letter 1]

14 Jan 2025

PONE-D-24-45387R1Sex and pressure effects of foam rolling on acute range of motion in the hamstring musclesPLOS ONE

Dear Dr. Hirose,

Thank you for submitting your manuscript to PLOS ONE. After careful consideration, we feel that it has merit but does not fully meet PLOS ONE’s publication criteria as it currently stands. Therefore, we invite you to submit a revised version of the manuscript that addresses the points raised during the review process.

**ACADEMIC EDITOR: **Dear Authors, your manuscript has been re-reviewed by one expert in the field that reported some minor issues you should consider during the revision process.

We look forward to receiving your revised manuscript.

Kind regards,

Emiliano Cè, Ph.D.

Academic Editor

PLOS ONE

Journal Requirements:

Additional Editor Comments (if provided):

Reviewers' comments:

Reviewer's Responses to Questions

**Comments to the Author**

1. If the authors have adequately addressed your comments raised in a previous round of review and you feel that this manuscript is now acceptable for publication, you may indicate that here to bypass the “Comments to the Author” section, enter your conflict of interest statement in the “Confidential to Editor” section, and submit your "Accept" recommendation.

Reviewer #2: All comments have been addressed

2. Is the manuscript technically sound, and do the data support the conclusions?

Reviewer #2: Yes

3. Has the statistical analysis been performed appropriately and rigorously? 

Reviewer #2: Yes

4. Have the authors made all data underlying the findings in their manuscript fully available?

Reviewer #2: Yes

5. Is the manuscript presented in an intelligible fashion and written in standard English?

Reviewer #2: Yes

6. Review Comments to the Author

Reviewer #2: Congratulations! I believe that the paper is stronger in its current form right now. I have only one concern regarding the participants section. In order to make it more clear and to avoid any confusion, can you please correct your statement in Line 93 "resulting in a total of 20 participants for each sex”? If you say 20 participants for each sex, it means 40 participants. So, you need to make it more straightforward to avoid any confusion.

7. PLOS authors have the option to publish the peer review history of their article (what does this mean? ). If published, this will include your full peer review and any attached files.

**Do you want your identity to be public for this peer review?** For information about this choice, including consent withdrawal, please see our Privacy Policy .

Reviewer #2: No

---

## [Author Response · Author response to Decision Letter 2]

15 Jan 2025

Response to Reviewer

Dear Dr. Emiliano Cè and Reviewers

Thank you for your valuable feedback and constructive comments on our manuscript. We appreciate the time and effort you have dedicated to reviewing our work and providing insightful suggestions. Below, we address your concern regarding the clarity of the participants section, specifically the statement in Line 93, "resulting in a total of 20 participants for each sex."

Comment:

The statement "resulting in a total of 20 participants for each sex" could lead to confusion, as it may imply 40 participants in total. Please revise to ensure clarity.

Response:

Thank you for pointing out this potential ambiguity. To make the statement clearer and avoid any misunderstanding, we have revised the text as follows:

"However, to reduce the influence of individual differences and measurement errors, address uncertainties in effect size, mitigate the impact of missing data and adjustments for non-sphericity, and enhance the reproducibility of the experimental results, the study included 10 participants per group. This resulted in a total of 20 participants (10 males and 10 females)."(L89-93)

This revision explicitly states the total number of participants and their distribution by sex, ensuring clarity and removing any potential confusion.

We hope this revision satisfactorily addresses your concern. Please let us know if there are any further points you would like us to address.

---

## [Decision Letter · Decision Letter 2]

29 Jan 2025

Sex and pressure effects of foam rolling on acute range of motion in the hamstring muscles

PONE-D-24-45387R2

Dear Dr. Hirose,

We’re pleased to inform you that your manuscript has been judged scientifically suitable for publication and will be formally accepted for publication once it meets all outstanding technical requirements.

Kind regards,

Emiliano Cè, Ph.D.

Academic Editor

PLOS ONE

Additional Editor Comments (optional):

Reviewers' comments:

Reviewer's Responses to Questions

**Comments to the Author**

1. If the authors have adequately addressed your comments raised in a previous round of review and you feel that this manuscript is now acceptable for publication, you may indicate that here to bypass the “Comments to the Author” section, enter your conflict of interest statement in the “Confidential to Editor” section, and submit your "Accept" recommendation.

Reviewer #2: All comments have been addressed

2. Is the manuscript technically sound, and do the data support the conclusions?

Reviewer #2: Yes

3. Has the statistical analysis been performed appropriately and rigorously? 

Reviewer #2: Yes

4. Have the authors made all data underlying the findings in their manuscript fully available?

Reviewer #2: Yes

5. Is the manuscript presented in an intelligible fashion and written in standard English?

Reviewer #2: Yes

6. Review Comments to the Author

Reviewer #2: Thank you for giving me the opportunity to review this article. I congratulate authors for their efforts and hard work.

7. PLOS authors have the option to publish the peer review history of their article (what does this mean? ). If published, this will include your full peer review and any attached files.

**Do you want your identity to be public for this peer review?** For information about this choice, including consent withdrawal, please see our Privacy Policy .

Reviewer #2: No

---

## [Editor Report · Acceptance letter]

PONE-D-24-45387R2

PLOS ONE

Dear Dr. Hirose,

I'm pleased to inform you that your manuscript has been deemed suitable for publication in PLOS ONE. Congratulations! Your manuscript is now being handed over to our production team.

Kind regards,

on behalf of

Prof. Emiliano Cè

Academic Editor

PLOS ONE